# SPANN: Highly-efficient Billion-scale Approximate Nearest Neighbor Search

**Qi Chen[1, *]   Bing Zhao[1, 2, †]   Haidong Wang[1]   Mingqin Li[1]   Chuanjie Liu[1, 3, †]**
**Zengzhong Li[1]   Mao Yang[1]   Jingdong Wang[1, 4, *, †]**
[1]Microsoft     [2]Peking University     [3]Tencent     [4]Baidu
[1]{cheqi, haidwa, mingqli, jasol, maoyang}@microsoft.com
[2]its.bingzhao@pku.edu.cn    [3]liu.chuanjie@outlook.com    [4]wangjingdong@outlook.com

## Abstract

The in-memory algorithms for approximate nearest neighbor search (ANNS) have achieved great success for fast high-recall search, but are extremely expensive when handling very large scale database. Thus, there is an increasing request for the hybrid ANNS solutions with small memory and inexpensive solid-state drive (SSD). In this paper, we present a simple but efficient memory-disk hybrid indexing and search system, named SPANN, that follows the inverted index methodology. It stores the centroid points of the posting lists in the memory and the large posting lists in the disk. We guarantee both disk-access efficiency (low latency) and high recall by effectively reducing the disk-access number and retrieving high-quality posting lists. In the index-building stage, we adopt a hierarchical balanced clustering algorithm to balance the length of posting lists and augment the posting list by adding the points in the closure of the corresponding clusters. In the search stage, we use a query-aware scheme to dynamically prune the access of unnecessary posting lists. Experiment results demonstrate that SPANN is $2\times$ faster than the state-of-the-art ANNS solution DiskANN to reach the same recall quality $90\%$ with same memory cost in three billion-scale datasets. It can reach $90\%$ recall@1 and recall@10 in just around one millisecond with only about $10\%$ of original memory cost. Code is available at: https://github.com/microsoft/SPTAG.

billion-scale, vector search, inverted index solution

## 1   Introduction

Vector nearest neighbor search has played an important role in information retrieval area, such as multimedia search and web search, which provides relevant results by searching vectors with minimum distance to the query vector. Exact solutions for K-nearest neighbor search [49, 40] are not applicable in big data scenario due to substantial computation cost and high query latency. Therefore, researchers have proposed many kinds of approximate nearest neighbor search (ANNS) algorithms in the literature [11, 18, 38, 10, 14, 31, 34, 13, 29, 21, 16, 26, 43, 42, 33, 44, 37, 32, 19, 27, 9, 12, 39, 50, 20, 36]. However, most of the algorithms mainly focus on how to do low latency and high recall search all in memory with offline pre-built indexes. When targeting to the super large scale vector search scenarios, such as web search, the memory cost will become extremely expensive. There is an increasing request for the hybrid ANNS solutions that use small memory and inexpensive disk to serve the large scale datasets.

---

*Corresponding author.
†Work done while at Microsoft.

35th Conference on Neural Information Processing Systems (NeurIPS 2021).

There are only a few approaches working on the hybrid ANNS solutions, including DiskANN [39] and HM-ANN [36]. Both of them are graph based solutions. DiskANN uses Product Quantization (PQ) [25] to compress the vectors stored in the memory while putting the navigating spread-out graph along with the full-precision vectors on the disk. When a query comes, it traverses the graph according to the distance of quantized vectors and then reranks the candidates according to distance of the full-precision vectors. HM-ANN leverages the heterogeneous memory by placing pivot points in the fast memory and navigable small world graph in the slow memory without data compression. However, it consumes more than 1.5 times larger fast memory than DiskANN. Moreover, the slow memory is still much expensive than disk. Therefore, due to the cheap serving cost, high recall and low latency advantages of DiskANN, it has become the start-of-the-art for indexing billion-scale datasets.

In this paper, we argue that the simple inverted index approach can also achieve state-of-the-art performance for large scale datasets in terms of recall, latency and memory cost. We propose SPANN, a simple but surprising efficient memory-disk hybrid vector indexing and search system, that follows the inverted index methodology. SPANN only stores the centroid points of the posting lists in the memory while putting the large posting lists in the disk. We guarantee both low latency and high recall by greatly reducing the number of disk accesses and improving the quality of posting lists. In the index-building stage, we use a hierarchical balanced clustering method to balance the length of posting lists and expand the posting list by adding the points in the closure of the corresponding clusters. In the search stage, we use a query-aware scheme to dynamically prune the access of unnecessary posting lists. Experiment results demonstrate that SPANN is more than two times faster than the state-of-the-art disk-based ANNS algorithm DiskANN to reach the same recall quality 90% with same memory cost in three billion-scale datasets. It can reach 90% recall@1 and recall@10 in just around one millisecond with only about 10% of original memory cost. SPANN has already been deployed into Microsoft Bing to support hundreds of billions scale vector search.

## 2 Background and Related Work

Given a set of data vectors $\mathbf{X} \in \mathbb{R}^{n \times m}$ (the data set contains $n$ vectors with $m$-dimensional features) and a query vector $\mathbf{q} \in \mathbb{R}^m$, the goal of vector search is to find a vector $\mathbf{p}^*$ from $\mathbf{X}$, called nearest neighbor, such that $\mathbf{p}^* = \arg\min_{\mathbf{p} \in \mathbf{X}} \mathrm{Dist}(\mathbf{p}, \mathbf{q})$. Similarly, we can define $K$-nearest neighbors. Due to the substantial computation cost and high query latency of the exhaustive search, ANNS algorithms are designed to speedup the search for the approximate $K$-nearest neighbors in a large dataset in an acceptable amount of time. Most of the ANNS algorithms in the literature mainly focus on the fast high-recall search in the memory, including hash based methods [14, 24, 47, 48, 45, 46, 51], tree based methods [11, 31, 44, 33], graph based methods [21, 16, 43, 32], and hybrid methods [42, 12, 23, 22]. However, with the explosive growth of the vector scale, the memory has become the bottleneck to support large scale vector search. There are only a few approaches working on the ANNS solutions for billon-scale datasets to minimize the memory cost. They can be divided into two categories: inverted index based and graph based methods.

The inverted index based methods, such as IVFADC [26], FAISS [27] and IVFOADC+G+P [9], split the vector space into $K$ Voronoi regions by KMeans clustering and only do search in a few regions that are closed to the query. To reduce the memory cost, they use vector quantization, e.g. Product Quantization (PQ) [25], to compress the vectors and store them in the memory. The inverted multi-index (IMI) [7] also uses PQ to compress vectors. It splits the feature space into multiple orthogonal subspaces and constructs a separate codebook for each subspace. The full feature space is produced as a Cartesian product of the corresponding subspaces. Multi-LOPQ[28] uses locally optimized PQ codebook to encode the displacements in the IMI structure. GNO-IMI [8] optimizes the IMI by using non-orthogonal codebooks to produce the centroids. Although they can cut down the memory usage to less than 64GB for one billion 128 dimensional vectors, the recall@1 is very low (only around 60%) due to lossy data compression. Although they can achieve better recall by returning 10 to 100 times more candidates for further reranking, it is often not acceptable in many scenarios.

The graph based methods include DiskANN [39] and HM-ANN [36]. Both of them adopt the hybrid solution. DiskANN also stores the PQ compressed vectors in the memory while storing the navigating spread-out graph along with the full-precision vectors on the disk. When a query comes, it traverses the graph using best-first manner according to the distance of quantized vectors and then reranks

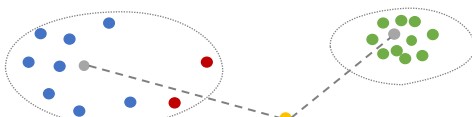

Figure 1: Example of boundary vector missing due to partial search. If we search yellow point, we will search green posting list first since the centroid of green posting list is closer to the yellow point although there are some boundary points (colored red) in the blue posting list that are much closer.

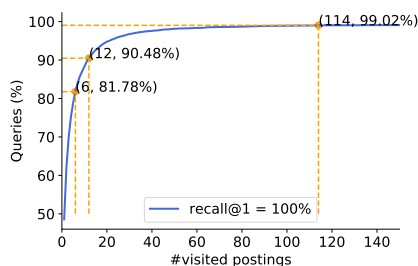

Figure 2: Different queries require different number of posting lists for search. To recall top one result on SIFT1M dataset, we find 80% of queries only need to search 6 posting lists, while 99% of queries need to search 114 posting lists.

the candidates according to distance of the full-precision vectors. Similarly, it uses the lossy data compression which will influence the recall quality even though full-precision vector reranking can help retrieve some missing candidates back. The high-cost random disk accesses limit the number of graph traverse and candidate reranking. HM-ANN leverages the heterogeneous memory by placing pivot points promoted by the bottom-up phase in the fast memory and navigable small world graph in the slow memory without data compression. However, it will lead to more than 1.5 times larger fast memory consumption. Moreover, the slow memory is still much expensive than disk and may be not available in some platforms. The theoretical analysis of the limits and the benefits of the graph based methods are given in [35].

## 3 SPANN

In this paper, we propose SPANN, a simple but efficient vector indexing and search system, that follows the inverted index methodology. Different from previous inverted index based methods that leverage the lossy data compression to reduce the memory cost, SPANN adopts a simple memory-disk hybrid solution.

**Index structure**: The data vectors $\mathbf{X}$ are divided into $N$ posting lists $\{\mathbf{X}_1, \mathbf{X}_2, \cdots, \mathbf{X}_N\}$, $\mathbf{X}_1 \cup \mathbf{X}_2 \cup ... \cup \mathbf{X}_N = \mathbf{X}^3$. The centroids of these posting lists, $\mathbf{c}_1, \mathbf{c}_2, \cdots, \mathbf{c}_N$, are stored in the memory as the fast coarse-grained index that point to the location of the corresponding posting lists in the disk.

**Partial search**: When a query $\mathbf{q}$ comes, we find the $K$ closest centroids, $\{\mathbf{c}_{i1}, \mathbf{c}_{i2}, \ldots, \mathbf{c}_{iK}\}$, $K \ll N$, and load the vectors in the posting lists $\mathbf{X}_{i_1}, \mathbf{X}_{i_2}, \cdots, \mathbf{X}_{i_K}$ that correspond to the closest $K$ centroids into memory for further fine-grained search.

### 3.1 Challenges

**Posting length limitation:** Since all the posting lists are stored in the disk, in order to reduce the disk accesses, we need to bound the length of each posting list so that it can be loaded into memory in only a few disk reads. This requires us to not only partition the data into a large number of posting lists but also balance the length of posting lists. This is very difficult due to the substantial high clustering cost and the balance partition problem itself. The imbalanced posting lists will lead to high variance of query latency especially when posting lists are stored in the disk.

**Boundary issue:** The nearest neighbor vectors of a query $\mathbf{q}$ may locate in the boundary of multiple posting lists. Since we only search a small number of relevant posting lists, some true neighbors of $\mathbf{q}$ that located in other posting lists will be missing (Illustrated in Figure 1). If red points are only represented by the centroid of blue posting list, they will be missing in the nearest neighbor search of yellow point.

---

[3]For convenience, we use $\mathbf{X}$ to denote both the matrix and the vector set.

**Diverse search difficulty:** We find that different queries may have different search difficulty. Some queries only need to be searched in one or two posting lists while some queries require to be searched in a large number of posting lists (Illustrated in Figure 2). If we search the same number of posting lists for all queries, it will result in either low recall or long latency.

All of the above challenges are the reasons why all of previous inverted index approaches adopt lossy data compression solution that stores all the compressed vectors and the posting lists in the memory.

### 3.2 Key techniques to address the challenges

In this paper, we introduce three key techniques that solve the above challenges to enable the memory-disk hybrid solution. In the index-building stage, we firstly limit the length of the posting lists to effectively reduce the number of disk accesses for each posting list in the online search. Then we improves the quality of the posting list by expanding the points in the closure of the corresponding posting lists. This increases the recall probability of the vectors located on the boundary of the posting lists. In the search stage, we propose a query-aware scheme to dynamically prune the access of unnecessary posting lists to ensure both high recall and low latency. The detail design of each technique will be introduced in the following sections.

#### 3.2.1 Posting length limitation

Limiting the length of posting lists means we need to partition the data vectors $\mathbf{X}$ into a large number of posting lists $\mathbf{X}_1, \mathbf{X}_2, \cdots, \mathbf{X}_N$. Balancing the length of posting lists means we need to minimize the variance of the posting length $\sum_{i=1}^{N}(|\mathbf{X}_i| - |\mathbf{X}|/N)^2$.

To address the posting length balance problem, we can leverage the multi-constraint balanced clustering algorithm [30] to partition the vectors evenly into multiple posting lists:

$$\min_{\mathbf{H},\mathbf{C}} ||\mathbf{X} - \mathbf{H}\mathbf{C}||_{\mathrm{F}}^2 + \lambda \sum_{i=1}^{N}(\sum_{l=1}^{|\mathbf{X}|} h_{li} - |\mathbf{X}|/N)^2, \text{ s.t. } \sum_{i=1}^{N} h_{li} = 1. \tag{1}$$

where $\mathbf{C} \in \mathbb{R}^{N \times m}$ is the centroids, $\mathbf{H} \in \{0, 1\}^{|\mathbf{X}| \times N}$ represents the cluster assignment, $\sum_{l=1}^{|\mathbf{X}|} h_{li}$ is the number of vectors assigned to the $i$-th cluster (i.e. $|\mathbf{X}_i|$) and $\lambda$ is a trade-off hyper parameter between clustering and balance constraints.

However, we find that when the vector number $|\mathbf{X}|$ and the partition number $N$ are very large, directly using multi-constraint balanced clustering algorithm cannot work due to the difficulty of large $N$-partition balanced clustering problem and the extremely high clustering cost. Therefore, we introduce a hierarchical multi-constraint balanced clustering technique (Figure 3) to not only reduce the clustering time complexity from $O(|\mathbf{X}| * m * N)$ to $O(|\mathbf{X}| * m * k * log_k(N))$ ($k$ is a small constant) but also balance the length of posting lists. We cluster the vectors into a small number (i.e. $k$) of clusters iteratively until each posting list contains limit number of vectors. By using this technique, we can greatly reduce not only the length of each posting list (disk accesses) but also the index build cost.

Moreover, since the number of centroids is very large, finding the nearest posting lists for a query needs to consume large computation cost. In order to make the navigating computation more meaningful, we replace the centroid with the vector that is closest to the centroid to represent each posting list. Then the wasted navigating computation is transformed to the distance computation for a subset of real candidates.

What's more, in order to quickly find a small number of nearest posting lists for a query, we create a memory SPTAG [12] (MIT license) index for all the vectors that represent the centorids of the posting lists. SPTAG constructs space partition trees and a relative neighborhood graph as the vector index which can speedup the nearest centroids search to sub-millisecond response time.

#### 3.2.2 Posting list expansion

To deal with boundary issue, we need to increase the visibility for those vectors that are located in the boundary of the posting lists. One simple way is to assign each vector to multiple close clusters. However, it will increase the posting size significantly leading to the heavy disk reads. Therefore, we

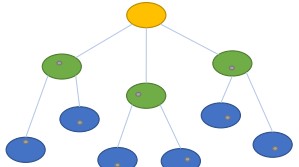 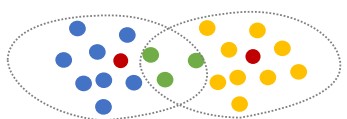 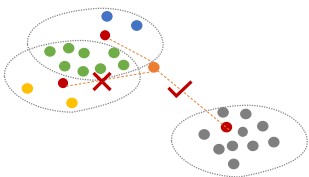

Figure 3: Hierarchical balanced clustering: iteratively balanced partition the vectors in a large cluster (yellow cluster) into a small number of small clusters (green clusters) until each cluster only contains limit number of vectors (blue clusters).

Figure 4: Closure clustering assignment: assign boundary vectors (green points) to multiple closest clusters if its distances to these clusters are nearly the same (blue and yellow clusters).

Figure 5: Representative replication: use RNG rule to reduce the similarity of two close posting lists. The orange point will be assigned to blue and grey posting lists although it is closer to yellow list than grey list.

introduce a closure multi-cluster assignment solution for boundary vectors on the final clustering step – assign a vector to multiple closest clusters instead of only the closest one if the distance between the vector and these clusters are nearly the same (Figure 4 gives an example):

$$\mathbf{x} \in \mathbf{X}_{ij} \iff \mathrm{Dist}(\mathbf{x}, \mathbf{c}_{ij}) \leq (1 + \epsilon_1) \times \mathrm{Dist}(\mathbf{x}, \mathbf{c}_{i1}),$$
$$\mathrm{Dist}(\mathbf{x}, \mathbf{c}_{i1}) \leq \mathrm{Dist}(\mathbf{x}, \mathbf{c}_{i2}) \leq \cdots \leq \mathrm{Dist}(\mathbf{x}, \mathbf{c}_{iK}) \tag{2}$$

This means we only duplicate the boundary vectors. For those vectors which are very close to the centroid of a cluster, they still keep one copy. By doing so, we can effectively limit the capacity increase due to closure cluster assignment while increasing the recall probability of these boundary vectors: they will be recalled if any of their closest posting lists is searched.

Since each posting list is small and we use closure assignment which will result in some posting lists that are very close to each other contain the same duplicated vectors (For example, the green vectors belong to both yellow and blue clusters). Too many duplicated vectors in the close posting lists will also waste the high-cost disk reads. Therefore, we further optimize the closure clustering assignment by using RNG rule [41] to choose multiple representative clusters for the assignment of an boundary vector in order to reduce the similarity of two close posting lists (Figure 5). RNG rule can be simply defined as we will skip the cluster $ij$ for vector $\mathbf{x}$ if $\mathrm{Dist}(\mathbf{c}_{ij}, \mathbf{x}) > \mathrm{Dist}(\mathbf{c}_{ij-1}, \mathbf{c}_{ij})$. The insight is two close posting lists are more likely to be both recalled by the navigating index. Instead of storing similar vectors in close posting lists, it would be better to store different vectors to increase the number of seen vectors in the online search. From the vector side, it is better to be represented by posting lists located in different directions (blue and grey posting lists in the example) than just being represented by posting lists located in the same direction (blue and yellow posting lists).

### 3.2.3 Query-aware dynamic pruning

In the index-search stage, to process different queries effectively with different resource budget, we introduce the query-aware dynamic pruning technique to reduce the number of posting lists to be searched according to the distance between query and centroids. Instead of searching closest $K$ posting lists for all queries, we dynamically decide a posting list to be searched only if the distance between its centroid and query is almost the same as the distance between query and the closest centroid:

$$\mathbf{q} \overset{search}{\iff} \mathbf{X}_{ij} \iff \mathrm{Dist}(\mathbf{q}, \mathbf{c}_{ij}) \leq (1 + \epsilon_2) \times \mathrm{Dist}(\mathbf{q}, \mathbf{c}_{i1}),$$
$$\mathrm{Dist}(\mathbf{q}, \mathbf{c}_{i1}) \leq \mathrm{Dist}(\mathbf{q}, \mathbf{c}_{i2}) \leq \cdots \leq \mathrm{Dist}(\mathbf{q}, \mathbf{c}_{iK}) \tag{3}$$

By further reducing those unnecessary posting lists in the closest $K$ posting lists, we can significantly reduce the query latency while still preserving the high recall by leveraging the resource more reasonably and effectively.

## 4 Experiment

In this section we first present the experimental comparison of SPANN with the current state-of-the-art ANNS algorithms. Then we conduct the ablation studies to further analyze the contribution of each

technique. Finally, we setup an experiment to demonstrate the scalability of SPANN solution in the distributed search scenario.

## 4.1 Experiment setup

We conduct all the experiments on a workstation machine with Ubuntu 16.04.6 LTS, which is equipped with two Intel Xeon 8171M CPU (2600 MHz frequency and 52 CPU cores), 128GB memory and 2.6TB SSD organized in RAID-0. The datasets we use are as follows:

1. SIFT1M dataset [3] is the most commonly used dataset generated from images for evaluating the performance of memory-based ANNS algorithms, which contains one million of 128-dimensional float SIFT descriptors as the base set and 10,000 query descriptors as the test set.

2. SIFT1B dataset [3] is a classical dataset for evaluating the performance of ANNS algorithms that support large scale vector search, which contains one billion of 128-dimensional byte vectors as the base set and 10,000 query vectors as the test set.

3. DEEP1B dataset [8] is a dataset learned from deep image classification model which contains one billion of 96-dimensional float vectors as the base set and 10,000 query vectors as the test set.

4. SPACEV1B dataset [6](O-UDA license) is a dataset from commercial search engine which derives from production data. It represents another different content encoding – deep natural language encoding. It contains one billion of 100-dimensional byte vectors as a base set and 29,316 query vectors as the test set.

The comparison metrics to demonstrate the performance are:

1. Recall: We compare the $R$ vector ids returned by ANNS with the $R$ ground truth vector ids to calculate the recall@$R$. Since there exist multiple data vectors sharing the same distance with the query vector, we also replace some of the ground truth vector ids with the vector ids that sharing the same distance to the query vector in the recall calculation.

2. Latency: We use the query response time in milliseconds as the query latency.

3. VQ (Vector-Query): The product of the number of vectors and the number of queries per second a machine can serve (which is introduced in GRIP [50]). It demonstrates the serving capacity of the search engine which takes both query latency and memory cost into consideration. The higher VQ the system has, the less resource cost it consumes. Here we use the number of vectors per KB $\times$ the number of queries per second as the VQ capacity.

## 4.2 SPANN on single machine

In this section, we demonstrate that inverted index based SPANN solution can also achieve the state-of-the-art performance in terms of recall, latency and memory cost. We first compare SPANN with the state-of-the-art billion-scale disk-based ANNS algorithms on three billion-scale datasets. Then we conduct an experiment on SIFT1M dataset to compare the VQ capacity with the start-of-the-art all-in-memory ANNS algorithms. For all the experiments in this section, we use the following hyper-parameters for SPANN: 1) use at most 8 closure replicas for each vector in the closure clustering assignment; 2) limit the max posting list size to 12KB for byte vectors and 48KB for float vectors; 3) set $\epsilon_1$ for posting list expansion to 10.0, and set $\epsilon_2$ for query-aware dynamic pruning with recall@1 and recall@10 to 0.6 and 7.0 , respectively. We increase the maximum number of posting lists to be searched in order to get the different recall quality.

### 4.2.1 Comparison with state-of-the-art billion-scale disk-based ANNS algorithms

We choose the state-of-the-art disk-based ANNS algorithms that can support billion-scale datasets as our comparison targets. we do not compare with HM-ANN [36] since it is not open sourced and the required PMM hardware may not be available in some platforms. Therefore, we compare SPANN only with the state-of-the-art billion-scale disk-based ANNS algorithm DiskANN. We use the default hyper parameters for DiskANN (same as the paper [39] for SIFT1B and SPACEV1B, and the pre-build index provided by [2] for DEEP1B).

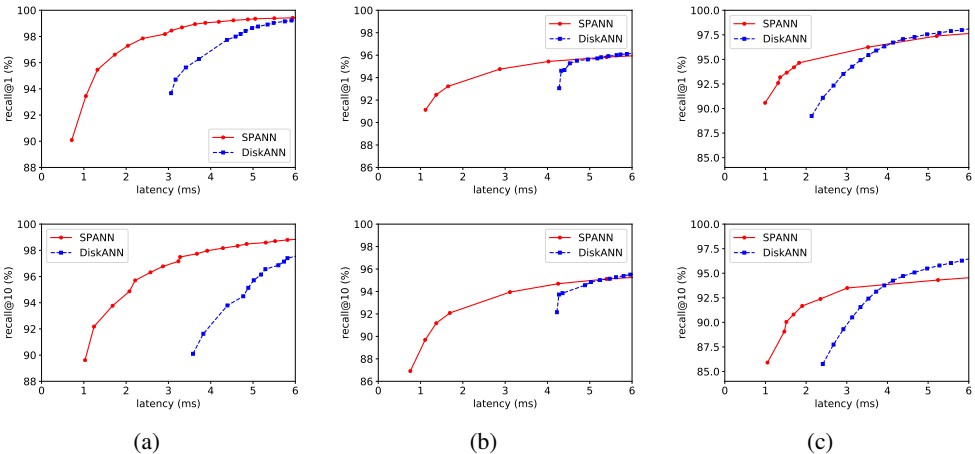

Figure 6: SPANN vs. DiskANN on (a) SIFT1B, (b) SPACEV1B, and (c) DEEP1B.

We carefully adjust the navigating memory index size of SPANN by choosing suitable number of posting lists (about 10-12% of total vector number) to ensure both the algorithms consume the same amount of memory (about 32GB for SIFT1B and SPACEV1B datasets and 60GB for DEEP1B dataset). Figure 6(a) demonstrates the performance for SIFT1B dataset. From the results, we find SPANN significantly outperforms DiskANN in both recall1@1 and recall@10 especially in the low query latency budget (less than 4ms). Especially, SPANN is more than two times faster than DiskANN to reach the 95% recall@1 and recall@10.

The performance results for SPACEV1B and DEEP1B datasets are shown in Figure 6(b) and 6(c). It also demonstrates that SPANN outperforms DiskANN in both recall@1 and recall@10 when query latency budget is small (less than 4ms). Especially, DiskANN cannot achieve good recall quality (90%) in less than 4ms in SPACEV1B dataset, while SPANN can obtain a recall of 90% in just around 1ms. For DEEP1B dataset, SPANN can also be more than two times faster than DiskANN to reach the good recall quality (90%).

### 4.2.2 Comparison with state-of-the-art all-in-memory ANNS algorithms

Then we conduct an experiment on SIFT1M dataset to compare the VQ capacity with the start-of-the-art all-in-memory ANNS algorithms, NSG [19], HNSW [32], SCANN [20], NGT-ONNG [23], NGT-PANNG [22] and N2 [4]. These algorithms have presented state-of-the-art performance in the ann-benchmarks [1]. We choose VQ capacity instead of latency as the comparison metric since these algorithms use much more memory to trade for low latency. However, memory is an expensive resource which has become the bottleneck for those algorithms to support large scale datasets. Therefore, we should take both memory and latency into consideration in the performance comparison. We take the SIFT1M dataset as an example due to the memory bottleneck of our test machine. We believe that the observation can be generalized to billion scale datasets.

Most of these algorithms are graph based algorithms. For NSG, we get the pre-built index from [5] and run the performance test with varying SEARCH_L from 1 to 256 which controls the quality of the search results. For HNSW (nmslib), SCANN, NGT-ONNG, NGT-PANNG and N2 we use the hyper parameters they provided in the ann-benchmarks [1] that achieve the best performance for the SIFT1M dataset.

Figure 7 and 8 demonstrate the VQ capacity and query latency of all the algorithms on recall@1 and recall@10. We can see from the result that SPANN achieves the best VQ capacity consistently across almost all the recall levels. This means although SPANN cannot achieve as low latency as the all-in-memory ANNS algorithms due to the high-cost disk accesses during the search, it can obtain the best serving capacity in the large scale vector search scenario.

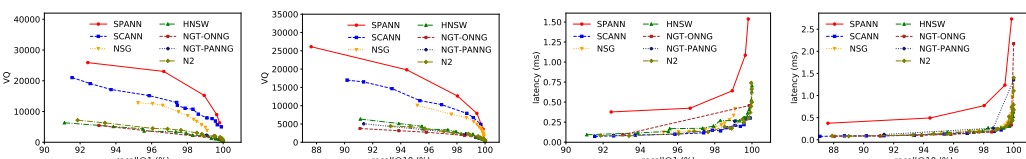

Figure 7: VQ of different ANNS indices          Figure 8: Latency of different ANNS indices

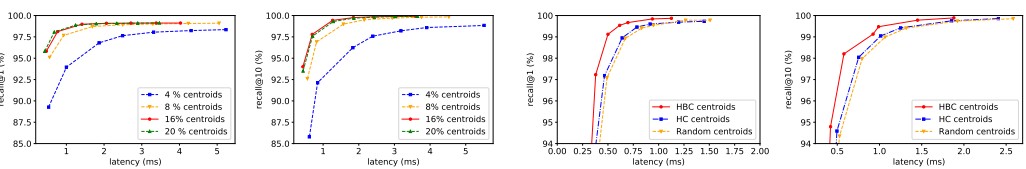

Figure 9: Different numbers of centroids          Figure 10: Different types of centroid selection

### 4.2.3 Ablation studies

In this section, we conduct a set of experiments to do the ablation studies on each of our techniques in the SIFT1M dataset.

**Hierarchical balanced clustering** There are three fast ways to partition the vectors on a single machine into a large number of posting lists: 1) randomly choose a set of points as the posting list centroids; 2) using hierarchical KMeans clustering (HC) to select centroids; 3) using hierarchical balanced clustering (HBC) to generate a set of centroids. We compare the index quality by generating 16% points as the centroids using these three ways.

Figure 10 shows the recall and latency performance of these three centroid selection algorithms. For both recall@1 and recall@10, we can see HBC centroid selection is better than random and HC selections, which demonstrates that balance posting length is very important for inverted index based methods. Moreover, HBC is very fast which clusters one million points into 160K clusters in only around 50 seconds with 64 threads. The whole SPANN index can be built in around 2 minutes.

Moreover, how many centroids are needed? Small number of centroids can reduce the navigating memory index size. However, large number of centroids usually means better performance. Therefore, we need to make reasonable trade-off between the memory usage and the performance. Figure 9 compares the performance of different numbers of centroids. From the result, we can see that the performance will increase significantly with the growth of the centroid number when the centroid number is small. However, when the number of centroids becomes large enough (16%), the performance will not increase any more. Therefore, we can choose 16% of points as the centroids to achieve both good search performance and small memory usage.

**Closure clustering assignment** To use closure clustering assignment, we need to assign a vector to multiple closed clusters to increase its recall probability during the search. Then at most how many closure replicas we need to duplicate for a vector to ensure the performance? Too small replicas cannot help to retrieve those boundary vectors back. However, too many replicas will increase the posting size greatly which will also affect the performance. Figure 11 demonstrates the performance of different numbers of replicas for closure clustering assignment. From the result, we can see that using more than one replicas improves the performance significantly. However, when the number of replicas is larger than 8, the performance cannot be improved any more. Therefore, we choose 8 replicas for all of our experiments.

**Query-aware dynamic pruning** In order to process different queries effectively during the online search, we introduce the query-aware dynamic pruning technique to further reduce the number of posting lists to be searched by pruning those unnecessary posting lists in the closest $K$ posting lists. We compare the performance with and without query-aware dynamic pruning in the Figure 12. From the result, we can see that with query-aware dynamic pruning, we can further reduce the query latency without recall drop especially when the latency budget is small. Note that, this technique can reduce not only the query latency but also the resource usage for a query.

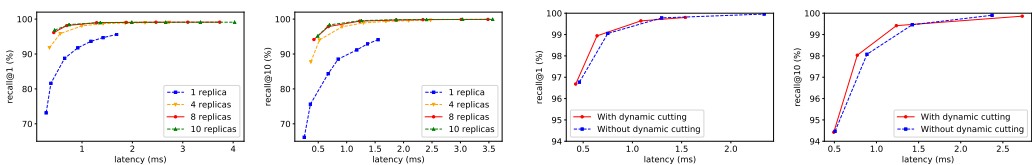

Figure 11: Different numbers of closure replicas     Figure 12: W/O query-aware dynamic pruning

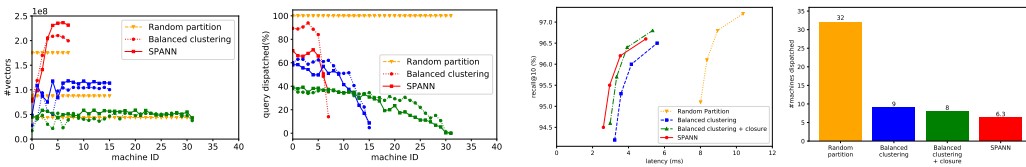

Figure 13: Data size and query access distribution across different machines     Figure 14: Comparison of recall, latency and #machines dispatched in the end-to-end test

## 4.3 Extension of SPANN to distributed search scenario

Compared to the graph base approaches, the additional advantage for inverted index based SPANN approach is that the partial search on the nearest posting lists idea can be easily extended to the distributed search scenario, which can handle super large scale vector search with high efficiency and low serving cost. The approach Pyramid [15] demonstrates the power of balanced partition and partial search approach in the distributed scenario. To demonstrate the scalability of SPANN in distributed search scenario, we partition the data vectors $\mathbf{X}$ evenly into $M$ partitions $\{\mathbf{X}_1, \mathbf{X}_2, \cdots, \mathbf{X}_M\}$ by using the multi-constraint balanced clustering and closure clustering assignment techniques in the distributed index build stage, where $M$ is the number of machines. In the online search stage, we also adopt the query-aware dynamic pruning technique to reduce the number of dispatched machines, which effectively limits the total cpu and IO cost for a query.

The only challenge for us is that there may have some hot-spot machines. Therefore, we need to balance not only the data size but also the query access in each machine to avoid the hot spots. To address the hot-spot challenge, we partition the vectors into multiple small partitions (larger than machine number) and then use best-fit bin-packing algorithm [17] to pack the small partitions into large bins (the number of bins equals to the number of machines) according to the history query access distribution. By doing so, we can effectively balance not only the data size but also the queries processed on each machine.

We compare the optimized SPANN solution with traditional random partition and all dispatch solution to demonstrate the effectiveness of workload reduction and scalability of SPANN in distributed search scenario. We conduct the experiments below based on the SPACEV1B dataset and use about 100,000 query accesses history from production as the test workload. The workload is evenly split into three sets: train, valid and test. The train set is used in offline distributed index build, and the test set is used in the online search evaluation.

### 4.3.1 Workload reduction and scalability

Figure 13 shows the number of vectors and the number of test query accesses in each machine when partitioning all the base vectors into 8, 16, and 32 partitions. From the result, we can see that SPANN distributes all the data and query accesses evenly into different machines. Although it increases the number of vectors in each machine by 20% due to closure assignment, it significantly reduces the query accesses in each machine compared to the random partition solution. Moreover, SPANN can continually reduce the query accesses in each machine by using more machines while random partition cannot. This means we can always add more machines to support more queries per second, which demonstrates good scalability of our system. The reason why we can achieve good scalability is that we effectively bound the number of machines to do the search for each query.

#### 4.3.2 Analysis

Then we analyze how each technique affects the performance. We use 32 partitions case to do the ablation study. We build SPANN single machine index for each partition and use the 29,316 query vectors with ground truth as the test workload. Figure 14 demonstrates the recall, latency and the average number of machines to dispatch in the end-to-end distributed search scenario. The left figure shows that SPANN can achieve almost the best recall in each latency budget. In the right figure, we can see that random partition solution needs to dispatch the query to all 32 machines for search. Using multi-constraints balanced clustering technique can significantly reduce the number of dispatched machines to 9. By adding closure assignment, we can further reduce the number of dispatched machines to 8. When all the techniques applied (including query-aware dynamic pruning in the online search), we can finally reduce the number of dispatched machines to 6.3. This means we can save about 80.3% of computation and IO cost for a query. Meanwhile, by reducing the number of machines to search for a query, we can further reduce the query latency since we reduce the number of candidates for final aggregation.

## 5 Conclusion

In this paper, we introduce SPANN, a simple but surprising efficient inverted index based ANNS system, which achieves state-of-the-art performance for large scale datasets in terms of recall, latency and memory cost. Different from previous inverted index based methods that use lossy data compression to address the memory bottleneck, SPANN adopts a simple memory-disk hybrid solution which only stores the centroids of the posting lists in the memory. We guarantee both low latency and high recall by greatly reducing the number of disk accesses and improving the quality of posting lists. Experiment results show SPANN can not only establish the new state-of-the-art performance for billion scale datasets but also achieve good scalability when extended to distributed search scenario. This demonstrates the ability of hierarchical SPANN to support super large scale vector search with high efficiency and low serving cost.

## 6 Acknowledgements

We would like to thank Ben Karsin and Murat Guney from Nvidia to help us further speedup the SPANN index build with GPU, which runs more than five times faster than CPU version.

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
