# OpenReview forum: "SPANN: Highly-efficient Billion-scale Approximate Nearest Neighborhood Search"
_NeurIPS.cc/2021/Conference — NeurIPS 2021 Spotlight_

### Official Review · Reviewer_2vRb · 2021-07-14

**Rating:** 8
**Confidence:** 4

**Summary:**

This paper proposes SPANN, which uses the inverted index to conduct large-scale nearest neighbor search on SSD. The centroids are kept in memory and indexed by a neighborhood graph for fast access. The lists of vectors associated with the centroids are stored on disk. Optimizations include (1) hierarchical balanced clustering to balance and limit the sizes of the lists; (2) assigning vectors on cluster boundaries to multiple clusters; (3) adjusting the number of lists to check according to each query. Experiment results show that SPANN significantly outperforms DiskANN, a state-of-the-art SSD-based solution, and is effective for distributed search.




**Limitations And Societal Impact:**

Yes

**Main Review:**

I like that the optimizations are reasonable, and the performance of SPANN is good. I think the experiment design and presentation clarity can be significantly improved.

1. In Eq. (2) and Eq. (3), do the two \epsilon take the same value? How do you set or tune them? Assigning a vector to at most 8 clusters and checking at most L lists for a query are only worst-case controls and do not tell exactly how \epsilon is configured.

2. In Section 4.2.2, when comparing with state-of-the-art ANN algorithms, SPANN stores only the centroids or all data in memory? For this experiment, it is OK to report the VQ capacity but recall-time curve must be reported as it is most widely recognized performance metric for similarity search.

3. Please report the index construction cost of SPANN on SIFT1B and SPAVEC1B. I am curious to which degree replicating vectors across clusters blow up the index size.

4. Comparing with randomly sampled clusters in Figure 9 and random partition in Figure 13 and 14 are not reasonable as the random baselines apparently will not work well. To show the benefits of hierarchical balanced clustering, you may compare with hierarchical k-means without cluster size control.

5. For distributed similarity search, [1] has shown that clustering-based data partitioning can reduce the number of accessed machines for each query and should be discussed. For distributed similarity search, please report a recall-time performance comparison with the baselines to show the end-to-end performance benefits of SPANN.

6. According to Figure 11 and Figure 12, it seems that the main performance gain of SPANN comes from multiple cluster assignment while the benefit of dynamic pruning is marginal.

7. Both HNSW[29] and N2[30] refers to HNSW (an archive version and a journal version), and they are treated as two different algorithms in the experiments. What are the differences between them?

8. Presentation can be significantly improved. “nearest neighborhood search” should be “nearest neighbor search”. “In order to make the result reproducible, we conduct the experiments below based on the SPACEV1B dataset.” Why experiments on SIFT1B will not be reproducible?

[1] Pyramid: A General Framework for Distributed Similarity Search on Large-scale Datasets

**Time Spent Reviewing:**

4

---

> ### Author Response · Authors · 2021-08-09
> **The response for the major concerns**
>
> Thanks to the reviewers for the insightful suggestions! We address the major concerns below:
>
> **Q1: In Eq. (2) and Eq. (3), do the two \epsilon take the same value? How do you set or tune them? Assigning a vector to at most 8 clusters and checking at most L lists for a query are only worst-case controls and do not tell exactly how \epsilon is configured.**
>
> **R1:** Thank you very much for your comments. The two \epsilon in Eq. (2) and Eq. (3) are not the same (sorry for the confusion due to the same symbol, we will use different symbols in our final version).
>
> We tuned the two \epsilon values based on the SIFT1M dataset and then directly applied to the SIFT1B and SPACEV1B datasets. The \epsilon we use for posting list expansion is 10.0. The \epsilon we use for query-aware dynamic pruning for recall@1 is 0.6. The \epsilon we use for query-aware dynamic pruning for recall@10 is 7.0.
>
> We believe that tuning the two variables directly on the large datasets with a validation set will lead to better results. We will add the tuning curves for the two \epsilon in the final version.
>
> **Q2: In Section 4.2.2, when comparing with state-of-the-art ANN algorithms, SPANN stores only the centroids or all data in memory? For this experiment, it is OK to report the VQ capacity but recall-time curve must be reported as it is most widely recognized performance metric for similarity search.**
>
> **R2:** Thank you very much for your suggestions. Yes, when comparing with state-of-the-art all-in-memory ANNS algorithms, SPANN stores only the centroids in memory while putting the posting lists in the disk. We want to clarify that our goal is not to apply our approach for small datasets that the small memory, such as 128G, is enough. Instead, we would like to demonstrate SPANN can achieve highest VQ capacity in the billion scale datasets that memory cost is very expensive for all-in-memory algorithms. Unfortunately, we cannot build billion-scale indexes for all-in-memory algorithms due to the memory bottleneck of our test machine. Therefore, we use a smaller dataset as an example to demonstrate the VQ capacity of SPANN and all-in-memory algorithms. We think that the observation can be generalized to billion scale datasets.
>
> We report the recall-latency number of all these algorithms as follows. From the results we can see that compared to the best all-in-memory ANNS algorithm, SPANN is about 5.4X and 6.1X slower (1.2X and 1.2X better VQ capacity) to reach 90% recall@1 and recall@10, 4.2X and 4.0X slower (1.5X and 1.7X better VQ capacity) to reach 95% recall@1 and recall@10, 3.5X and 5.6X slower (1.9X and 1.2X better VQ capacity) to reach 99% recall@1 and recall@10. We will add the recall-latency curves in the final version.
>
> Algorithms| Recall@1 (%) | Latency (ms)
> :--:|:--:|:--:
> SPANN| [92.43, 96.68, 98.94, 99.64, 99.80] |  [0.38, 0.42, 0.64, 1.09, 1.54]
> SCANN| [92.56, 97.40, 99.09, 99.67, 99.90] |  [0.08, 0.12, 0.21, 0.29, 0.30]
> NSG  | [92.97, 96.62, 98.79, 99.01, 99.10] |  [0.11, 0.13, 0.28, 0.33, 0.42]
> HNSW | [91.12, 97.35, 99.14, 99.62, 99.85] |  [0.09, 0.17, 0.27, 0.31, 0.41]
> NGT-ONNG | [81.96, 93.05, 99.85] |  [0.05, 0.09, 0.46]
> NGT-PANNG | [83.18, 99.07] |  [0.06, 0.26]
> N2 | [91.86, 96.89, 98.91, 99.56, 99.83] |  [0.08, 0.14, 0.17, 0.24, 0.30]
>
> Algorithms | Recall@10 (%) | Latency (ms)
> :--:|:--:|:--:
> SPANN | [87.62, 94.42, 98.03, 99.42, 99.86] |  [0.37, 0.49, 0.77, 1.24, 2.73]
> SCANN | [87.08, 93.38, 98.70, 99.16, 99.70] |  [0.08, 0.10, 0.19, 0.22, 0.31]
> NSG   | [95.18, 97.60, 98.67, 99.43, 99.88] |  [0.16, 0.21, 0.24, 0.33, 0.67]
> HNSW  | [91.12, 95.51, 97.93, 99.34, 99.85] |  [0.09, 0.13, 0.20, 0.27, 0.41]
> NGT-ONNG | [91.10, 93.86, 95.82, 98.53, 99.85] |  [0.09, 0.10, 0.12, 0.18, 0.33]
> NGT-PANNG | [82.30, 91.36, 98.63, 99.99] |  [0.08, 0.12, 0.26, 1.35]
> N2 | [87.90, 94.67, 97.73, 99.50, 99.87] |  [0.09, 0.14, 0.18, 0.30, 0.48]
>
> **Q3: Please report the index construction cost of SPANN on SIFT1B and SPAVEC1B. I am curious to which degree replicating vectors across clusters blow up the index size.**
>
> **R3:** Thank you very much for your suggestions. For the index build time cost, using CPU with 45 threads to build the billion-scale indexes in parallel, SPANN needs 4.1 and 5.1 days to build the SIFT1B and SPACEV1B indexes, and DiskANN needs 3.5 and 4.2 days. Fortunately, our approach SPANN is GPU-friendly: using 4 V100 GPU cards, we can reduce the index build time to only 1.2 day. According to our experience, graph-based algorithm is not easy to fully leverage the gpu to speedup the index build.
>
> For the index size, the disk cost of SPANN and DiskANN is comparable: the index size of SPANN and DiskANN in SPACEV1B dataset is 821.89GB and 930.57GB, in SIFT1B dataset is 833.87GB and 667.47GB. The average replica number of the vectors in SPANN is 4.5 and 5.1 for SPACEV1B and SIFT1B.
>
> **Q4: Comparing with randomly sampled clusters in Figure 9 and random partition in Figure 13 and 14 are not reasonable as the random baselines apparently will not work well. To show the benefits of hierarchical balanced clustering, you may compare with hierarchical k-means without cluster size control.**
>
> **R4:** Thank you very much for your valuable suggestions. Regarding "Comparing with randomly sampled clusters in Figure 9",
> we followed your advice and conducted the evaluation on the hierarchical k-means without cluster size control (HC)
> which is almost the same as randomly sampled centroids (we will add it into figure 9 in the final version). This might show that posting length balance and limitation is more important than center quality.
>
> For figure 13 and 14 in distributed setting, in the real business products (hundreds of billion scale), it is often required that all the partitions have roughly the same data size to maximize the resource utilization.
> K-means without cluster size control usually cannot meet this requirement.
>
> We have provided a strong baseline (balanced clustering) in figure 14 which just applied the balanced clustering technique without closure multi-cluster assignment and query-aware dynamic pruning. From figure 14, we can see that SPANN solution can further reduce 30% of the computation and IO cost as well as the query aggregation latency. For figure 13, we will add the comparison with the balanced clustering baseline in the final version.
>
> **Q5: For distributed similarity search, [1] has shown that clustering-based data partitioning can reduce the number of accessed machines for each query and should be discussed. For distributed similarity search, please report a recall-time performance comparison with the baselines to show the end-to-end performance benefits of SPANN.**
>
> **R5:** Thank you very much for your suggestions. We will follow your advice to discuss [1] and state that
> clustering-based data partitioning in [1] and our balanced clustering enjoy similar roles for reducing the number of accessed machines.
>
> We want to point out that other techniques in SPANN: balance not only the data size but also the query accesses, use closure assignment to improve the quality of the posting lists, and use query-aware dynamic pruning to further prune unnecessary machines, can further reduce the number of accessed machines effectively (We think you already notice the effect of these techniques).
>
> **Q6: For distributed similarity search, please report a recall-time performance comparison with the baselines to show the end-to-end performance benefits of SPANN.**
>
> **R6:** Thank you very much for your valuable suggestions. The end-to-end recall-time performance comparison of SPANN with the baselines (we also use the single machine SPANN to build the ANNS indexes for all partitions) are as follows (The result of balanced clustering is still running, we will report it as soon as the result is ready):
>
> Solutions | recall@10 (%) | Latency (ms)
>  :--:|:--:|:--:
> Random partition | [0.869, 0.896, 0.9208, 0.939439] |  [8.01, 8.36, 8.95, 10.37]
> Balanced clustering + closure assignment | [0.932545, 0.944341, 0.951375, 0.956048] |  [2.98, 3.34, 3.92, 5.34]
> SPANN | [0.931856, 0.943621, 0.950614, 0.955253] |  [2.60, 2.95, 3.54, 4.96]
>
> From the results, we can see that "SPANN" and "Balanced clustering + closure assignment" can achieve better recall and lower latency than "Random partition". We think the reason might be that closure assignment increases the recall probability of boundary vectors. "SPANN" achieves almost the same recall with "Balanced clustering + closure assignment" while it effectively reduces the query latency due to the dynamic query-aware pruning.
>
> **Q7: According to Figure 11 and Figure 12, it seems that the main performance gain of SPANN comes from multiple cluster assignment while the benefit of dynamic pruning is marginal.**
>
> **R7:** Yes, for single machine SPANN, the performance gain of dynamic pruning is marginal compared with multiple cluster assignment. However, in the distributed setting, the performance gain of dynamic pruning is 1.7X larger than multiple cluster assignment (see figure 14).
>
> **Q8: Both HNSW[29] and N2[30] refers to HNSW (an archive version and a journal version), and they are treated as two different algorithms in the experiments. What are the differences between them?**
>
> **R8:** Thank you very much for your comments. We are sorry that we use the wrong reference for N2. The correct reference for N2 is https://github.com/kakao/n2. N2 is another implementation for HNSW. It achieves better query latency and less memory usage than HNSW (nmslib).
>
> **Q9: Presentation can be significantly improved. “nearest neighborhood search” should be “nearest neighbor search”. “In order to make the result reproducible, we conduct the experiments below based on the SPACEV1B dataset.” Why experiments on SIFT1B will not be reproducible?**
>
> **R9:** Thank you for pointing out this misleading description. We did not mean that the experiments on SIFT 1B will not be reproducible. We will modify the presentation.

---

> > ### Comment · Reviewer_2vRb · 2021-08-18
> > **Comments after rebuttal**
> >
> > I have read the rebuttal and the authors have addressed all my concerns. I will raise my rating to 8.

---

### Official Review · Reviewer_nUdz · 2021-07-14

**Rating:** 8
**Confidence:** 4

**Summary:**

SPANN is a hybrid disk/RAM vector indexing method.

SPANN uses a hierarchical clustering (with some tricks) to split the dataset into buckets that are stored uncompressed on disk. At search time, the buckets corresponding to the nearest centroids to the query (again with some tricks) are visited to retrieve the nearest neighbors.

The indexing-time tricks are intended to balance the bucket sizes and reduce assignment ambiguity with some level of multiple storage.

At search time, the number of buckets to visit is tuned based on the distance of the query to the centroids.

There is also a section about a distributed version of the method.

**Ethical Concerns:**

-

**Limitations And Societal Impact:**

-

**Main Review:**

Strengths

S1 The paper is very practical and has lots of interesting insights.

S2 results clearly outperform the state of the art.

S3 an attempt to normalize vector serving cost between disk-based and memory-only methods.

Weaknesses

W1 very engineering-oriented, no methodological insights

W2 The fairness of the VQ comparison is discussible

S2 There are not so many hybrid RAM/disk indexing methods. DiskANN is one of them and is a good comparison point. It may be worthwhile to compare the index building times, which tends to be slow for DiskANN

S3/W2. The VQ comparison measure between disk-based and memory-only methods is very interesting. However, the paper misses a self-contained description on how it works exactly. The SIFT1M dataset is so small that it easily fits in the 128G RAM of the test machine, including with graph structure overheads. So it seems that the disk capacity and RAM are wasted, meaning that a smaller machine could be used.

W1 Especially the section about the distributed setting is largely orthogonal to the method, it could be applied similarly to other inverted-list based techniques. Including it makes that the paper looks more like a "systems" paper than a NeurIPS research paper.

There are a few questions that are left after reading the paper:

1. since the number of inverted lists to visit depends on the query, how much variation is there in the query time?

2. the authors mention that each vector is duplicated at most 8 times, but what is total number of duplicated vectors that need to be stored?

3. since the search from different inverted lists can yield several times the same vector, some over-retrieval and deduplication needs to be applied. It may be useful to discuss that.




**Time Spent Reviewing:**

1.5

---

> ### Author Response · Authors · 2021-08-09
> **The response for the major concerns**
>
> Thanks to the reviewers for the insightful suggestions! We address the major concerns below:
>
> **Q1: very engineering-oriented, no methodological insights.**
>
> **R1:** Yes, we might not have new methodological insights in the concept level. Our main contribution lies in how to turn the concept-level ideas into
> workable algorithm specifically for memory-disk hybrid vector search in large ANNS scenario that achieves state-of-the-art performance in terms of recall, latency and memory cost.
>
> **Q2: The fairness of the VQ comparison is discussible. The VQ comparison measure between disk-based and memory-only methods is very interesting. However, the paper misses a self-contained description on how it works exactly. The SIFT1M dataset is so small that it easily fits in the 128G RAM of the test machine, including with graph structure overheads. So it seems that the disk capacity and RAM are wasted, meaning that a smaller machine could be used.**
>
> **R2:** Thank you very much for your suggestions. Yes, you are right that SIFT1M dataset does not need 128G RAM and large disk. Our goal is not to apply our approach
> for small datasets that the small memory, such as 128G, is enough.
> Instead, we would like to demonstrate SPANN can achieve highest VQ capacity in the billion scale datasets that memory cost is very expensive for all-in-memory algorithms. Unfortunately, we cannot build billion-scale indexes for all-in-memory algorithms due to the memory bottleneck of our test machine. Therefore, we use a smaller dataset as an example to demonstrate the VQ capacity of SPANN and all-in-memory algorithms. We think that the observation can be generalized to billion scale datasets.
>
> **Q3: It may be worthwhile to compare the index building times, which tends to be slow for DiskANN.**
>
> **R3:** Thank you very much for your suggestions. In our machine, using CPU with 45 threads to build the billion-scale indexes in parallel, SPANN needs 4.1 and 5.1 days to build the SIFT1B and SPACEV1B indexes, and DiskANN needs 3.5 and 4.2 days. Fortunately, our approach SPANN is GPU-friendly: using 4 V100 GPU cards, we can reduce the index build time to only 1.2 day. According to our experience, graph-based algorithm is not easy to fully leverage the gpu to speedup the index build.
>
> **Q4: Especially the section about the distributed setting is largely orthogonal to the method, it could be applied similarly to other inverted-list based techniques. Including it makes that the paper looks more like a "systems" paper than a NeurIPS research paper.**
>
> **R4:** Thank you very much for your comments. It is true that all the inverted index based techniques can be applied to the distributed setting. This is not our point. Our point is to demonstrate that the three ideas: posting length limitation, posting list expansion and query-aware dynamic pruning can also work well in the distributed setting with a small modification.
>
> By applying these three ideas, we can greatly reduce the total cpu and IO cost for a query while still achieving high recall. Meanwhile, the query latency can also be improved since the number of candidates for final aggregation is effectively cut down.
>
> **Q5: since the number of inverted lists to visit depends on the query, how much variation is there in the query time?**
>
> **R5:** Thank you very much for your suggestions.
>
> We report the query latency (query time) distribution at 50%, 90%, 95% and 99% percentile for the two datasets to reach 90% recall,
> which is more useful in real systems,
> to characterize the variation.
> The results are as the following:
>
> SPACEV1B|Average latency (ms)|50% percentile latency (ms)|90% percentile latency (ms)|95% percentile latency (ms)|99% percentile latency (ms)
> :--:|:--:|:--:|:--:|:--:|:--:
> recall@1|1.117|1.059|1.407|1.519|2.315
> recall@10|1.109|1.066|1.352|1.454|2.067
>
> SIFT1B|Average latency (ms)|50% percentile latency (ms)|90% percentile latency (ms)|95% percentile latency (ms)|99% percentile latency (ms)
> :--:|:--:|:--:|:--:|:--:|:--:
> recall@1|0.714|0.716|0.787|0.808|0.854
> recall@10|1.029|1.001|1.214|1.255|1.328
>
> You can see from the two tables that the query latency at 99% percentile is only about 1.2 to 2.0 times larger than the average query latency.
>
> **Q6: the authors mention that each vector is duplicated at most 8 times, but what is total number of duplicated vectors that need to be stored?**
>
> **R6:** The average replica number of the vectors in SPANN is 4.5 and 5.1 for SPACEV1B and SIFT1B.
> BTW, besides the disk cost from the replica,
> there are also some disk space being used to pad the posting list in order to make the posting list 4K aligned for fast disk read.
>
> **Q7: since the search from different inverted lists can yield several times the same vector, some over-retrieval and deduplication needs to be applied. It may be useful to discuss that.**
>
> **R7:** Thank you very much for your suggestions. Yes, a vector can be yielded several times from different inverted lists in the search. However, we only calculate the distance between query and a vector once. In SPANN, we save the vector along with its unified id in the posting lists. During search, we use an optimized hash map to record whether the distance between query and a vector id is computed or not for deduplication.

---

### Official Review · Reviewer_hs5r · 2021-07-14

**Rating:** 6
**Confidence:** 4

**Summary:**

The paper claims the following contributions. (1) It proposes a new disk-based algorithm (called SPANN) for billion-scale approximate nearest neighbor search on a single machine. (2) SPANN adopts two techniques. One is hierarchical balanced clustering for index building and the other is query-aware schema for search. (3) SPANN is two times faster than a state-of-the-art solution (called DiskANN).

**Limitations And Societal Impact:**

The paper studies the approximate nearest neighbor search problem which has been studied for decades. It seemed the paper does not introduce negative societal impact.

**Main Review:**

The paper studied a practical problem that aims at improving billion-scale Approximate Nearest Neighbor Search (ANNS). The adopted hierarchical balanced clustering technique is insightful. By copying the same vector into multiple posting lists, the proposed SPANN algorithm achieves faster query processing (see Figure 4 and 5 and Ablation Study). Experiments demonstrate that SPANN is about 2 times faster than the state-of-the-art DiskANN in query processing.

Despite the above strengths, the paper needs to fix a few issues to justify the claimed contributions adequately. Please see the following:

(1)	The paper claims that SPANN is faster than the state-of-the-art DiskANN. The claim is valid in terms of query processing time and according to Figure 6, query processing time of SPANN is about half of DiskANN when the recall is the same. However, the claim does not adequately support that SPANN is better than DiskANN because SPANN may use larger index size than DiskANN. It is trivial to improve DiskANN query processing time by using multiple indexes, which is a common practice for ANNS algorithms such as Locality Sensitive Hashing and Vector Quantization. The paper did not present index size in the results so it is still early to conclude that SPANN is better than DiskANN.

(2)	The paper claims a 2x faster performance improvement. The improvement seems marginal and there could be little statistical significance. It will be good if the paper can achieve a more significant improvement (e.g. 5x) on some datasets or include statistical significance tests on all datasets.

(3)	The paper did not claim technical contributions in the paper. Instead, the paper said it adopted or used existing techniques, i.e. hierarchical balanced clustering and query-aware schema. If the contribution is combining different techniques, the paper may add more techniques (e.g. 3 techniques). If the contribution is developing new techniques, the paper may justify how the proposed techniques differ from related papers (i.e. DiskANN, HNSW, and NSG).

[comments after author response]
I have increased my rating from 5 to 6.


**Time Spent Reviewing:**

5

---

> ### Author Response · Authors · 2021-08-09
> **The response for the major concerns**
>
> Thanks to the reviewers for the insightful suggestions! We address the major concerns below:
>
> **Q1: However, the claim does not adequately support that SPANN is better than DiskANN because SPANN may use larger index size than DiskANN. It is trivial to improve DiskANN query processing time by using multiple indexes, which is a common practice for ANNS algorithms such as Locality Sensitive Hashing and Vector Quantization. The paper did not present index size in the results so it is still early to conclude that SPANN is better than DiskANN.**
>
> **R1:** Your comment "SPANN may use larger index size than DiskANN" is not correct. In fact, the index sizes are comparable: the index size of SPANN and DiskANN in SPACEV1B dataset is 821.89GB and 930.57GB, in SIFT1B dataset is 833.87GB and 667.47GB. As a result, our claim “SPANN is better than DiskANN” holds.
>
> Our approach is related to multiple indexes, but is different from multiple indexes, such as LSH. In our approach, we build a single inverted index, and expand each posting list with closure assignment. In the typical multiple indexes approach, e.g. LSH, multiple tables (indexes) are built.
>
> **Q2: The paper claims a 2x faster performance improvement. The improvement seems marginal and there could be little statistical significance. It will be good if the paper can achieve a more significant improvement (e.g. 5x) on some datasets or include statistical significance tests on all datasets.**
>
> **R2:** Thank you very much for your suggestions. DiskANN is a strong baseline which represents the state-of-the-art performance for the large scale ANNS scenario. In fact, SPANN is 3.5X faster than DiskANN to reach recall 90% in SIFT1B dataset, 3X faster than DiskANN to reach recall 90% in SPACEV1B dataset.
>
> Such an improvement, though not reaching 5x, already has real values for our business products (hundreds of billion scale).
>
> We will add more evaluations on the datasets from more domains in the final version, such as speech, recommendation, etc.
>
> **Q3: The paper did not claim technical contributions in the paper. Instead, the paper said it adopted or used existing techniques, i.e. hierarchical balanced clustering and query-aware schema. If the contribution is combining different techniques, the paper may add more techniques (e.g. 3 techniques). If the contribution is developing new techniques, the paper may justify how the proposed techniques differ from related papers (i.e. DiskANN, HNSW, and NSG).**
>
> **R3:** Thank you very much for your comments. Our contribution is to propose a simple but efficient inverted index based memory-disk hybrid vector indexing and search system which achieves state-of-the-art performance for large scale datasets in terms of recall, latency and memory cost. We want to point out that each component in our approach seems not very novel, but it is not trivial to make them work well
> for memory-disk hybrid vector search. The three challenges, i.e. posting length limitation, boundary issue and diverse search difficulty, can not be well solved using exiting inverted-index methods in the large scale vector ANNS scenario. The typical inverted index based approaches adopt lossy data compression to achieve low memory and low latency goals by storing the compressed vectors and the posting lists all in the memory. Unfortunately, the recall@1 of them is very low (only around 60%). Although they can achieve better recall by returning 10 to 100 times more candidates for further reranking, the additional large number of random disk accesses for reranking are often not acceptable as the latency increases a lot.
>
> In SPANN, we guarantee both low latency and high recall by greatly reducing the number of disk accesses and improving the quality of posting lists. In the index-building stage, we use a hierarchical balanced clustering method to balance the length of posting lists and expand the posting list by adding the points in the closure of the corresponding clusters. In the search stage, we use a query-aware scheme to dynamically prune the access of unnecessary posting lists.

---

> > ### Comment · Reviewer_hs5r · 2021-08-18
> > **I am happy to see the paper accepted**
> >
> > Thank you for the author response. Maybe I am biased towards existing baselines so I requested more comparisons in the original review. Overall, I am happy to see this paper accepted and have increased my rating from 5 to 6.

---

### Official Review · Reviewer_tJVR · 2021-07-15

**Rating:** 6
**Confidence:** 4

**Summary:**

In this paper, the authors study the approximate nearest neighbor search (ANNS) problem and develop an inverted index based algorithm using both memory and disk in the searching to reduce the required amount of memory. The proposed algorithm first partitions the vectors into clusters and then stores in the memory the centroid of each cluster and the disk address of the vectors belonging to the cluster, or called posting lists. To limit the length of posting lists (the number of vectors in a cluster), the algorithm performs the clustering with an additional constraint of balancing the length of each posting list, as suggested in [27]. But the algorithm performs the clustering in a hierarchical way so that the computational cost can be improved. It further replaces a centroid by the closest vector and another index [10] to speed up the search. It also assigns boundary vectors, those close to multiple clusters, to the multi-clusters to reduce the number of posting lists accessed. An RNG rule [37] that assigns a vector to clusters at different directions is employed to avoid generation of similar clusters. Finally, during the query, only posting lists that are sufficiently close to the query compared to the closest posting list will be searched. Experimental evaluations have been conducted on two datasets of one billion to demonstrate the superior performance.

**Main Review:**

The problem of large-scale ANNS using small memory consumption is of significance in the machine learning community. The authors achieve to show that the inverted index approach can be helpful in designing a low-memory ANNS algorithm. They carefully employ/use multiple ideas to adapt the inverted index method to the memory-disk hybrid setting and save the memory cost. Each of the ideas is clearly explained with an illustrating figure, making the paper easy to follow. Experimental results validate to some degree that the developed algorithm is memory-efficient while keeping high recall and small latency. The algorithm can be easily extended to the distributed setting to further improve the scalability.

The ideas are simple and straightforward/standard. The authors make use of multiple existing ideas and the biggest contribution to me is to show that inverted index can be adapted in the low-memory/hybrid setting. This is not too surprising because usually only a few posting lists are accessed when processing a query, well suited to the low-memory setting. The theoretical understanding of the inverted index based algorithm is still lack, unlike graph-based algorithms [a]. Although I believe this paper will be eventually published, the authors will need to test the algorithm in more and diverse large-scale datasets (instead of only two datasets). This is because different datasets have different characteristics and an algorithm can have quite different performance in different datasets. Including more datasets enables us to better understand the behavior of the algorithm in datasets from different domains in practice. The writing of the experimental parts needs to be enhanced. See detailed comments.

Reference:

[a] Liudmila Prokhorenkova and Aleksandr Shekhovtsov. Graph-based Nearest Neighbor Search: From Practice to Theory. ICML2020.

Suggestions:
1.	In the ablation study for the pruning idea (Fig. 12), tuning the eps parameter, which controls the allowed distance between the query and a centroid, can give more information than only with or without pruning.
2.	There appears to be some space left, e.g., Fig. 14 can be compressed into half-spacing. Discussing the different types of ANNS algorithms including graph-based algorithms, inverted index algorithms and LSH algorithms can provide readers better context.
3.	The term “posting list” appears in Line 48 for the first time but are implicitly defined in Section 3. The definition should be moved earlier.
4.	For comparisons with all-in-memory algorithms, it’s best to provide the latency in addition to the VQ capacity. Although the authors acknowledge that SPANN has higher latency, it is not unknown how much higher.  The loss in latency can be important for choosing an indexing algorithm in practice.

Detailed comments:
-	Typos: Line 121: “hybrid solution that solve”
-	Line 269: “random choose”
-	Line 276: “how many centroids we need to generate?”
-	Figure 10 and 11: “Different number of …”
-	Line 319: The logic of the sentence below is weird and it does not make sense to me. “In order to make the result reproducible, we conduct the experiments below based on the SPACEV1B dataset.”
-	Line 339: “Only use …, we can …”

UPDATES AFTER REBUTTAL:

Thanks! In the rebuttal, the authors have addressed my concerns adequately. The authors will need to make it clear that the proposal algorithm works best for the large-data only, because as shown in the tables in the rebuttal, the latency is about 3 ~ 6 times slower than that of other state-of-the-art in-memory algorithms for the same recall. I would like to see more datasets from different domains tested and the authors promised to include them in the final version. I have updated my score from 5 to 6.

**Time Spent Reviewing:**

24

---

> ### Author Response · Authors · 2021-08-09
> **The response for the major concerns**
>
> Thanks to the reviewers for the insightful suggestions! We address the major concerns below:
>
> **Q1: The ideas are simple and straightforward/standard. The authors make use of multiple existing ideas and the biggest contribution to me is to show that inverted index can be adapted in the low-memory/hybrid setting. This is not too surprising because usually only a few posting lists are accessed when processing a query, well suited to the low-memory setting.**
>
> **R1:** We agree that using inverted index based method is a straightforward way to achieve low-memory goal. However, we would like to point out that, besides low memory, it is also very important to achieve high recall and low latency. It is not trivial for inverted index based methods to achieve all these three goals. The three challenges, i.e. posting length limitation, boundary issue and diverse search difficulty, can not be well solved using exiting inverted-index methods
> in the large scale vector ANNS scenario.
> The typical inverted index based approaches adopt lossy data compression to achieve low memory and low latency goals by storing the compressed vectors and the posting lists all in the memory. Unfortunately, the recall@1 of them is very low (only around 60%).
> Although they can achieve better recall by returning 10 to 100 times more candidates for further reranking, the additional large number of random disk accesses for reranking are often not acceptable
> as the latency increases a lot.
>
> **Q2: The authors will need to test the algorithm in more and diverse large-scale datasets (instead of only two datasets). This is because different datasets have different characteristics and an algorithm can have quite different performance in different datasets. Including more datasets enables us to better understand the behavior of the algorithm in datasets from different domains in practice.**
>
> **R2:** Thank you very much for your valuable suggestions. We would like to point out that the two datasets are diverse and from two different domains (Image and NLP). We will follow your advice and add more evaluations on the datasets from more domains in the final version, such as speech, recommendation, etc.
>
> **Q3: In the ablation study for the pruning idea (Fig. 12), tuning the eps parameter, which controls the allowed distance between the query and a centroid, can give more information than only with or without pruning.**
>
> **R3:** Thank you very much for your comments.
> The two \epsilon in Eq. (2) and Eq. (3) are not the same (sorry for the confusion due to the same symbol, we will use different symbols in our final version).
>
> We tuned the two \epsilon values based on the SIFT1M dataset and then directly applied to the SIFT1B and SPACEV1B datasets. The \epsilon we use for posting list expansion is 10.0. The \epsilon we use for query-aware dynamic pruning for recall@1 is 0.6. The \epsilon we use for query-aware dynamic pruning for recall@10 is 7.0.
>
> We believe that tuning the two variables directly on the large datasets with a validation set will lead to better results. We will add the tuning curves for the two \epsilon in the final version.
>
> **Q4: The theoretical understanding of the inverted index based algorithm is still lack, unlike graph-based algorithms [a].**
>
> **R4:** Thank you very much for your suggestions, we will add the discussion about [a] and study if we can borrow some ideas to perform the analysis for our approach.
>
> **Q5: There appears to be some space left, e.g., Fig. 14 can be compressed into half-spacing. Discussing the different types of ANNS algorithms including graph-based algorithms, inverted index algorithms and LSH algorithms can provide readers better context.**
>
> **R5:** Thank you very much for your suggestions, we will add more discussion about the different types of ANNS algorithms (including graph-based, inverted index based, tree-based and hash-based approaches) in the final version.
>
> **Q6: The term “posting list” appears in Line 48 for the first time but are implicitly defined in Section 3. The definition should be moved earlier.**
>
> **R6:** Thank you very much for your suggestions. We will move the definition of posting list to its first appearance.
>
> **Q7: For comparisons with all-in-memory algorithms, it’s best to provide the latency in addition to the VQ capacity. Although the authors acknowledge that SPANN has higher latency, it is not unknown how much higher. The loss in latency can be important for choosing an indexing algorithm in practice.**
>
> **R7:** Thank you very much for your suggestions. First, we want to clarify that our goal is not to apply our approach for small datasets that the small memory, such as 128G, is enough. Instead, we would like to demonstate SPANN can achieve highest VQ capacity in the billion scale datasets that memory cost is very expensive for all-in-memory algorithms. Unfortunately, we cannot build billion-scale indexes for all-in-memory algorithms due to the memory bottleneck of our test machine. Therefore, we use a smaller dataset as an example to demonstrate the VQ capacity of SPANN and all-in-memory algorithms. We think that the obseration can be generalized to billion scale datasets.
>
> We report the recall-latency number of all these algorithms as follows. From the results we can see that compared to the best all-in-memory ANNS algorithm, SPANN is about 5.4X and 6.1X slower (1.2X and 1.2X better VQ capacity) to reach 90% recall@1 and recall@10, 4.2X and 4.0X slower (1.5X and 1.7X better VQ capacity) to reach 95% recall@1 and recall@10, 3.5X and 5.6X slower (1.9X and 1.2X better VQ capacity) to reach 99% recall@1 and recall@10. We will add the recall-latency curves in the final version.
>
> Algorithms| Recall@1 (%) | Latency (ms)
> :--:|:--:|:--:
> SPANN| [92.43, 96.68, 98.94, 99.64, 99.80] |  [0.38, 0.42, 0.64, 1.09, 1.54]
> SCANN| [92.56, 97.40, 99.09, 99.67, 99.90] |  [0.08, 0.12, 0.21, 0.29, 0.30]
> NSG  | [92.97, 96.62, 98.79, 99.01, 99.10] |  [0.11, 0.13, 0.28, 0.33, 0.42]
> HNSW | [91.12, 97.35, 99.14, 99.62, 99.85] |  [0.09, 0.17, 0.27, 0.31, 0.41]
> NGT-ONNG | [81.96, 93.05, 99.85] |  [0.05, 0.09, 0.46]
> NGT-PANNG | [83.18, 99.07] |  [0.06, 0.26]
> N2 | [91.86, 96.89, 98.91, 99.56, 99.83] |  [0.08, 0.14, 0.17, 0.24, 0.30]
>
> Algorithms | Recall@10 (%) | Latency (ms)
> :--:|:--:|:--:
> SPANN | [87.62, 94.42, 98.03, 99.42, 99.86] |  [0.37, 0.49, 0.77, 1.24, 2.73]
> SCANN | [87.08, 93.38, 98.70, 99.16, 99.70] |  [0.08, 0.10, 0.19, 0.22, 0.31]
> NSG   | [95.18, 97.60, 98.67, 99.43, 99.88] |  [0.16, 0.21, 0.24, 0.33, 0.67]
> HNSW  | [91.12, 95.51, 97.93, 99.34, 99.85] |  [0.09, 0.13, 0.20, 0.27, 0.41]
> NGT-ONNG | [91.10, 93.86, 95.82, 98.53, 99.85] |  [0.09, 0.10, 0.12, 0.18, 0.33]
> NGT-PANNG | [82.30, 91.36, 98.63, 99.99] |  [0.08, 0.12, 0.26, 1.35]
> N2 | [87.90, 94.67, 97.73, 99.50, 99.87] |  [0.09, 0.14, 0.18, 0.30, 0.48]
>
> **Q8: The writing of the experimental parts needs to be enhanced. See detailed comments …**
>
> **R8:** Thank you very much for pointing out these typos. We will fix them and improve the writing of the experimental parts carefully in the final version.

---

### Decision · Program_Chairs · 2021-09-27

**Decision:**

Accept (Spotlight)

**Comment:**

The paper presents an empirically efficient algorithm for disk-scale similarity search. The algorithm offers significant (2x or higher) improvements over the state of the art. The reviewers’ opinion was that the techniques used in the algorithm were relatively standard, but they were carefully engineered and put together to obtain a practical software artifact. One concern was that the experiments have been performed on only two data sets, but the authors promised further experiments in the final version of the paper.